# Analysis of MicroRNA Cargo in Circulating Extracellular Vesicles from HIV-Infected Individuals with Pulmonary Hypertension

**DOI:** 10.3390/cells13110886

**Published:** 2024-05-21

**Authors:** Aatish Mahajan, Sumedha Gunewardena, Alison Morris, Matthias Clauss, Navneet K. Dhillon

**Affiliations:** 1Division of Pulmonary and Critical Care Medicine, Department of Medicine, University of Kansas Medical Center, Mail Stop 3007, 3901 Rainbow Blvd, Kansas City, KS 66160, USA; 2Department of Molecular & Integrative Physiology, University of Kansas Medical Center, Kansas City, KS 66160, USA; 3Department of Medicine, University of Pittsburgh School of Medicine, Pittsburgh, PA 15213, USA; morrisa@upmc.edu; 4Pulmonary, Critical Care, Sleep and Occupational Medicine, Indiana University School of Medicine, Indianapolis, IN 46202, USA

**Keywords:** extracellular RNA, pulmonary vascular disease, substance abuse, cocaine

## Abstract

The risk of developing pulmonary hypertension (PH) in people living with HIV is at least 300-fold higher than in the general population, and illicit drug use further potentiates the development of HIV-associated PH. The relevance of extracellular vesicles (EVs) containing both coding as well as non-coding RNAs in PH secondary to HIV infection and drug abuse is yet to be explored. We here compared the miRNA cargo of plasma-derived EVs from HIV-infected stimulant users with (HIV + Stimulants + PH) and without PH (HIV + Stimulants) using small RNA sequencing. The data were compared with 12 PH datasets available in the GEO database to identify potential candidate gene targets for differentially altered miRNAs using the following functional analysis tools: ingenuity pathway analysis (IPA), over-representation analysis (ORA), and gene set enrichment analysis (GSEA). MiRNAs involved in promoting cell proliferation and inhibition of intrinsic apoptotic signaling pathways were among the top upregulated miRNAs identified in EVs from the HIV + Stimulants + PH group compared to the HIV + Stimulants group. Alternatively, the downregulated miRNAs in the HIV + Stimulants + PH group suggested an association with the negative regulation of smooth muscle cell proliferation, IL-2 mediated signaling, and transmembrane receptor protein tyrosine kinase signaling pathways. The validation of significantly differentially expressed miRNAs in an independent set of HIV-infected (cocaine users and nondrug users) with and without PH confirmed the upregulation of miR-32-5p, 92-b-3p, and 301a-3p positively regulating cellular proliferation and downregulation of miR-5571, -4670 negatively regulating smooth muscle proliferation in EVs from HIV-PH patients. This increase in miR-301a-3p and decrease in miR-4670 were negatively correlated with the CD4 count and FEV1/FVC ratio, and positively correlated with viral load. Collectively, this data suggest the association of alterations in the miRNA cargo of circulating EVs with HIV-PH.

## 1. Introduction

The use of antiretroviral medication has become increasingly important in recent years as a means of efficiently suppressing HIV replication and extending the survival of people living with HIV (PLWH). However, longer lifespans of PLWH on highly active antiretroviral therapy (HAART) have led to an increased incidence of co-morbidities including pulmonary hypertension (PH). Pulmonary hypertension, which is defined by elevated pressure in the peripheral pulmonary arteries and failure of the right ventricle, is caused by vasoconstriction and progressive blockage of the distal pulmonary arteries [1]. The risk of developing pulmonary hypertension in PLWH is 300–6000-fold more than in individuals without HIV infection [2]. The reports on the impact of antiretroviral therapy on HIV-PH continue to be contradictory, and individuals who have HIV-PH often die from the illness [3,4], thereby making HIV-PH one of the most serious non-infectious consequences of HIV infection [5]. Further, illicit drug use increases HIV transmission, disease progression, and non-compliance with ART, particularly in older people with pulmonary hypertension This “double hit” increases the risk of HIV-associated pulmonary hypertension (HIV-PH) and right heart failure [2].

The possible association of HIV-associated PH and the use of illicit drugs has been extensively studied in our lab. Research has indicated that the pulmonary vascular remodeling linked to the development of PH is enhanced in HIV-infected individuals, Simian Immunodeficiency Virus-infected macaques, and HIV transgenic rats by a second hit of drugs of abuse [6,7]. One of the main causes of PH pathophysiology, which leads to endothelial apoptosis and smooth muscle proliferation is loss of or reduction in bone morphogenetic protein receptor *(BMPR)-2* expression [8]. We reported previously that cocaine increases the HIV-Tat, Nef, and gp120-driven decrease in the expression of BMPRs in pulmonary arterial smooth muscle cells (PASMCs) [7,9], and this increase corresponded with the activation of pro-proliferative *TGF-β* signaling, promoting hyper-proliferation of these cells [10,11]. Our previous reports further suggest translational repression of *BMPR-2* by miRNAs in cocaine and/or HIV-1 protein(s)-mediated smooth muscle hyperplasia [9]. We did not observe a decrease in *BMPR-2* at the mRNA levels in these cells or in HIV-transgenic rats and PLWH exposed to cocaine unlike other forms of PH [12,13].

Non-coding microRNAs have gained increasing attention as major regulators of gene expression in health and disease. These are protected from degradation by incorporation into extracellular vesicles (EVs). EVs carry various types of non-coding RNAs including pre- and mature-miRNAs, tRNAs, long non-coding RNAs, and circular RNAs; and changes in the levels of these in EVs have been associated with the pathologic processes of various diseases, including PH [5,14,15,16,17]. These EVs are actively excreted into the systemic circulation during disease conditions allowing for investigation of their role as mediators and markers of disease pathogenesis. Previous studies from our lab have shown that EVs released by HIV-infected monocyte-derived macrophages can potentiate pulmonary vascular endothelial injury and smooth muscle proliferation [5], leading to the development of cardio-pulmonary dysfunction [18]. We particularly reported that miRNA cargo in the EVs derived from HIV-infected macrophages in the presence of cocaine treatment can activate the proliferative *PI3K/AKT* signaling in pulmonary arterial smooth muscle cells [5]. In continuation of these initial cell-culture findings, we here now investigated the miRNA cargo of circulating EVs from PLWH with and without PH to identify unique EV-linked miRNAs associated with pulmonary hypertension in stimulant users living with HIV infection.

## 2. Material and Methods

### 2.1. Human Samples and Data Collection

The de-identified human plasma samples from PLWH abusing stimulants with (PLWH + Stimulants + PH) and without (PLWH + Stimulants) PH from the National Neuro-AIDS Tissue Consortium (NNTC) were used for small RNA seq analysis on plasma-derived EVs (*n* = 8). The stimulant users consisted of poly-drug users such as opioids, amphetamines, and/or cocaine users. For validation of selected miRNAs, plasma samples collected at the University of Pittsburgh HIV Cohort approved by the institutional review board were also used. Details on the inclusion/exclusion criteria have been published in previous reports [18,19,20]. A total of 36 samples [HIV-Uninfected non-drug users (*n* = 6), HIV-Uninfected cocaine users (Coc) (*n* = 6), people living with HIV non-drug users (PLWH) (*n* = 6), PLWH cocaine users (PLWH + Coc) (*n* = 6), PLWH with pulmonary hypertension (PLWH + PH) (*n* = 12, *n* = 6 non-drug users and *n* = 6 cocaine users)] were used for validation. The individuals included in the cocaine group were those who self-reported the use of cocaine (inhaled, crack, or intravenously) during the six months before their visit and had not previously used any other stimulants, opioids, or sedatives. Those who did not use drugs stated that they had never used illegal drugs of any kind. Among PLWH without PH (cocaine and non-drug users), *n* = 7 of 12 were hepatitis C positive, while the PLWH + PH group had *n* = 4 of 11 positives with the status of one unknown. Further uninfected cocaine users were mostly positive for Hepatitis C (4/5 and 1 unknown). About 80% of HIV-infected individuals were smokers with only 1–2 patients being diabetic across all the groups. The details on their demographics, use of antiretroviral therapy, viral load, plasma CD4+ T cell count, pulmonary function test, echocardiography, and other co-morbidities were collected previously [18,19,20]. PH was identified as pulmonary artery systolic pressure (PASP) greater than 40 mmHg, and it was not limited to the existence of Group I PH.

### 2.2. Isolation of Extracellular Vesicles

Extracellular vesicles were isolated from 500 µL of human EDTA plasma using the exoEasy kit (QIAGEN Inc., Germantown, MD, USA) as in our published findings [18]. In short, samples of plasma were thawed on ice and then centrifuged at 300× *g* for ten minutes at 4 °C. After discarding the pellet, the supernatant was centrifuged for 20 min at 4 °C at 3000× *g*. The larger vesicles were removed from the supernatant by centrifuging it again for 30 min at 4 °C at 10,000× *g*. Following the filtration of the resultant supernatant using a 0.45 µm filter, the filtrate was processed to isolate small EVs using an exoEasy kit (Qiagen, Germantown, MD, USA) according to the manufacturer’s instructions. The quantity and size of the EVs was determined with the Nanosight LM10 system (Malvern Panalytical, Malvern, UK). In addition, EVs were characterized using a Transmission Electron Microscope (TEM), and Western blotting using CD9, CD81, flotillin-1, and Alix antibodies as detailed in our previous study [18] and shown in Appendix A.

### 2.3. RNA Isolation from EVs

Before RNA separation from EVs, 1 µL (5 nM) of Cel-miR-39-3p miRNA [21] of *Caenorhabditis elegans*, which lacks homology with human miRNA (MSY0000010; (QIAGEN Inc., Germantown, MD, USA)), was spiked into 32 µg of EVs. Total RNA was then extracted from spiked EVs using the Qiagen miRNeasy Kit followed by measurement of RNA purity and concentration using a UV–vis spectrophotometer.

### 2.4. Small RNA Sequencing Analysis

To obtain an unbiased global profile of small RNA present in EVs, small RNA sequencing on RNA isolated from EVs was performed by Quick Biology (Monrovia, CA, USA). The analysis was performed in biological quadruplicates. After 3′ adapter removal and low-quality bases filtering using the Cutadapt software-version 1.18 (https://cutadapt.readthedocs.io/en/stable/, accessed on 26 April 2024) [22], the small RNA reads were mapped to the miRBase database (version 21) using the Bowtie alignment software (Bowtie version 1.2.2, https://bowtie-bio.sourceforge.net/index.shtml, accessed on 26 April 2024) [23]. Reads that were too short (<16 bp) were removed before alignment.

MiRNA abundance estimates were obtained using the QIASeq miRNA tools (QIAGEN Inc., Germantown, MD, USA). Between 3.7 and 5.4 million reads were obtained for each sample. Expression normalization and differential expression calculations were performed in DESeq2 software (DESeq2 version 1.38.3, https://bioconductor.org/packages/release/bioc/html/DESeq2.html, accessed on 26 April 2024) [24] to identify statistically significant differentially expressed miRNAs. DESeq2 employs a negative binomial generalized linear model (NB-GLM) for statistical calculations and works with minimal levels of biological replication. The significance *p*-values were adjusted for multiple hypotheses testing using the Benjamini and Hochberg method [25] establishing a false discovery rate (FDR) for each miRNA. MiRNAs with an absolute fold difference ≥ 1.5 and FDR ≤ 0.1 were considered significant for downstream analysis.

### 2.5. MiRNA Gene Interaction Networks

We used the Ingenuity Pathway Analysis software (IPA; Ingenuity Systems, www.ingenuity.com (last accessed on 15 June 2021)) to construct interaction networks between the perturbed miRNAs and their target genes. IPA is an online software tool that provides a comprehensive data repository of information on genes, and miRNAs together with their experimentally validated and predicted interactions and biological functions. Gene targets were identified as either experimentally observed, predicted with high confidence, or predicted with moderate confidence. We analyzed 12 datasets, publicly available in the GEO database (GSM3819897, GSE48149, GSE15197, GSE79786, GSE53408, GSE126262, GSE126262, GSE15197, GSE48149, GSE69416-1, GSE69416-2, GSE69416-3), associated with pulmonary hypertension, to identify potential candidate genes for our miRNAs. To be selected, genes had to be significantly differentially expressed (absolute fold-change ≥ 1.5 and *p*-value ≤ 0.05) concordantly in at least two of the datasets and not discordantly expressed in any of the other datasets. Proceeding from the understanding that miRNAs generally act as post-transcriptional gene suppressors, only those miRNA-mRNA targets with opposing differential expression patterns were reported. We used IPA’s Molecule Activity Predictor (MAP) tool to predict the activated state of PH explained by the differential expression pattern of miRNAs in our data and the expression pattern of their associated target genes.

### 2.6. MiRNA Enrichment Analysis

We used the miRNA enrichment analysis and Annotation Tool (MiEAA) software (miEAA 2.1, https://ccb-compute2.cs.uni-saarland.de/mieaa/, accessed on 26 April 2024) [26] for miRNA enrichment analysis and annotation. MiEAA was used to identify significantly over-represented functional categories and pathways (*p*-value ≤ 0.05) associated with the differentially expressed miRNAs. It was also used for gene (miRNA) set enrichment analysis (GSEA) [27]. MiEAA queries a multitude of databases in its analysis including, miRTarBase [28] for miRNA gene target information, gene ontology [29] for annotations derived over miRTarBase, KEGG [30] for pathways, MNDR [31] for miRNA-associated diseases, and miRWalk [32] to provide a plethora of information on the perturbed miRNAs. Upregulated and downregulated miRNAs were analyzed separately in the over-representation analysis. We used QuickGO (https://www.ebi.ac.uk/QuickGO/annotations, accessed on 26 April 2024) [33] to identify genes associated with significantly enriched gene ontology terms specifically relevant to our study (GO2001243, GO0042127, GO0048662, GO0007169, GO0038110) and IPA to establish putative interactions between the perturbed miRNAs and those genes with potential regulatory effects.

### 2.7. Validation of miRNAs Using Quantitative RT-PCR

The RNA isolated from EVs was used to prepare cDNAs specific to miRNAs using miScript II RT Kit (QIAGEN Inc., Germantown, MD, USA), according to the manufacturer’s instructions followed by qRT-PCR by using QuantiTect SYBR Green PCR Kit (QIAGEN Inc., Germantown, MD, USA).

### 2.8. Statistical Analysis

One-way ANOVA was used for statistical analysis, and for multiple comparisons, a post hoc Bonferroni correction was applied (Prism; GraphPad, La Jolla, CA, USA). After applying Bonferroni correction, the data were considered statistically significant when *p*-values were ≤0.05.

## 3. Results

### 3.1. Enrichment of miRNAs Positively Regulating Smooth Muscle Proliferation and Negatively Regulating Apoptosis in the Plasma-Derived EVs from HIV-Infected Individuals with Pulmonary Hypertension

The RNA-seq analysis of RNA from plasma-derived EVs of HIV-infected stimulant users (HIV + Stimulants) and HIV-infected stimulant users with PH (HIV + Stimulants + PH) revealed differential expression patterns of miRNAs in EVs between individuals with and without PH as shown in the heatmap of Figure 1A and Table 1. Seven clusters of gene sets were identified by unsupervised hierarchical clustering (UHC). Notably, miRNAs in clusters 1–3 were downregulated in the HIV + Stimulants + PH group in comparison to HIV + Stimulants and were found to be enriched in pathways bearing gene ontology (GO) functional terms (Table 1). These changes were related to interleukin-2-mediated signaling, negative regulation of smooth muscle cell proliferation, transmembrane receptor protein tyrosine kinase signaling pathway, and small ubiquitin-like protein (SUMO) polymer binding. In contrast, miRNAs in clusters 4–7 were upregulated in HIV + Stimulants + PH samples in comparison to HIV + Stimulants and were enriched in pathways such as regulation of cell proliferation, negative regulation of intrinsic apoptotic signaling pathway, and positive regulation of interleukin-12 secretion. Importantly, 13 EV-derived miRNAs from the GO functional pathways listed above were found at higher levels while 14 were observed at lower levels (FC > 2 and FDR < 0.05) when patients within the HIV + Stimulants + PH grouping were compared to the HIV Stimulants only group (Figure 1B).

We next performed functional over-representation analysis (ORA) of the identified differentially expressed miRNAs using the web-based application miEAA as described in the Section 2. We selected some of the PH-relevant pathways (enriched) from the annotations derived over miRTarBase (gene ontology). As shown in Figure 2A, the ORA of upregulated miRNAs in HIV-infected stimulators with PH compared to non-PH subjects showed notable enrichment of miRNA targets mapping to pathways such as positive regulation of cell proliferation and negative regulation of intrinsic apoptosis signaling among the top 19 most significant pathways. Some of the highly significant upregulated miRNAs were common to the positive regulation of cell proliferation [34] and negative regulation of intrinsic apoptosis signaling pathways. They included miR-373-3p, miR-9-5p, miR-192-5p, miR-148a-3p, miR-92b-3p, miR-301a-3p, and miR-203b-3p (Figure 2A). Additional miRNAs involved in the regulation of cell proliferation including miR-4458, miR-92b-3p, miR-301a-3p, miR-373-3p, and miR-148a-3p were either experimentally observed or predicted to target *TGFβR2* and/or *TNF* molecules (Appendix A). MiRNA-92b-3p in addition to *TNF* has been experimentally validated to target Krüppel-like factor 4 (*KLF4*), *BMPR2*, CCAAT enhancer-binding protein alpha (*CEBPA*) [35,36,37], while miR-4458 is known to target cyclin-dependent kinase 6 (*CDK6*), Neurofibromin 2 (*NF2*), CEBPA, PR domain zinc finger protein 1 (*PRDM1*), and Signal transducer and activator of transcription 3 (*STAT 3*) in addition to *TGFBR2* and *TNF*. All these molecules belong to the pathways significantly involved in PH pathogenesis [38,39,40,41]. The significantly higher levels of miRNAs in EVs from the HIV + Stimulants + PH group over-representing negative regulation of intrinsic apoptosis signaling based on experimental validation included miR-133a-3p targeting *BCL2*, *BCL2L1/L2*, *MCL1*, and *AKT1*, miR-4458 targeting *MMP9* and *BCL2L1*, and miR-451a and miR-1-3p targeting *BCL2* (Appendix A).

The ORA analysis of downregulated miRNAs in EVs from the HIV + Stimulants + PH group had significant enrichment of interleukin-2 mediated signaling, transmembrane receptor protein kinase signaling, and negative regulation of smooth muscle cell proliferation among the top 25 most significant pathways (enrichment *p*-value < 0.05) (Figure 2B). Some of the common miRNAs in these pathways included miR-107, miR-3194-3p, miR-6778-3p, miR-19b-3p, miR-4670-3p, miR-5571-5p, and miR-6859-5p. Target genes of miRNAs in the interleukin-2-mediated signaling pathway included *IL2RG*, *IL2RB*, and *IL2RA*, whereas in transmembrane receptor protein tyrosine kinase signaling pathway Erb-B2 Receptor Tyrosine Kinase 4 (*ERRB4*), *ERBB3*, and cluster of differentiation 4 (*CD4*), *CD7*, SH2B Adaptor Protein 1 (*SH2B1*) were a few of the important targets (Appendix A). Targets associated with negative regulation of smooth muscle cell proliferation included Heme Oxygenase 1 (*HMOX1*), Insulin-like growth factor-binding protein 3 (*IGFBP3*), TNF Alpha Induced Protein 3 (*TNFAIP3*), and *BMPR2* (Appendix A). Overall, sequence analysis of miRNA cargo of EVs from the HIV + Stimulants + PH group suggests enrichment of miRNAs associated with factors related to pulmonary hypertension.

### 3.2. Interactions of Significantly Altered miRNA in EVs with Pulmonary Hypertension Associated Gene Profiles

To construct interaction networks between the perturbed miRNAs and their target genes, IPA analysis was performed on significantly altered miRNAs (absolute fold difference ≥ 1.5 and FDR ≤ 0.1) and their target genes involved in pulmonary hypertension. As shown in Figure 3A, we analyzed 12 datasets, available in the GEO database associated with PH, to identify potential candidate PH-associated gene targets for altered miRNAs. Known interactions of the miRNAs from EVs with these PH-associated genes revealed that a majority of these target genes were associated with more than one miRNA (Figure 3A). For example, 6phosphofructo-2-kinase/fructose-2,6-bisphosphatase 3 (*PFKBP3*) involved in glycolysis is a target of six significantly altered miRNAs in EVs whereas Membrane Spanning 4-Domains A4A (*MS4A4A*) which belongs to tumor-associated macrophages is a target of four EV-linked miRNAs.

We observed more association of genes with the miRNAs that were downregulated (Figure 3A) which includes the miR-216a-3p/miR-128 cluster predicted to activate Adenosine A2b Receptor (*ADORA2B*) [42], miR-185-3p leading to activation of *PFKFB3*, and miR-4520-3p involved in the activation of Solute Carrier Family 9 Member A1 (*SLC9A1*). The observed downregulated miR-19a-3p is predicted to activate the antiangiogenic thrombospondin 1 [43] and the PH-associated Membrane Spanning 4-Domains A4A (*MS4A4A*) Also, our data demonstrated upregulated microRNA, namely the miR-6764-5p/1915 cluster, which was predicted to inhibit Interleukin 18 Receptor Accessory Protein (IL-18RAP). This alters *IL-18* mediated pro-inflammatory signaling involved in pulmonary vascular remodeling [44].

We also used IPA’s Molecule Activity Predictor (MAP) to predict the activated state of pulmonary hypertension explained by the differential expression pattern of miRNAs in our data and the expression pattern of their associated target genes. As illustrated in Figure 3B, the IPA-MAP tool predicted the inhibition of *BMPR2* by mature miR-32-5p and by another miRNA from the same family: miR-92b-3p [45], both of which were upregulated in EVs from the HIV + Stimulants + PH group. Along with this, indirect activation of TNF Superfamily Member 10 (*TNFSF10*) was predicted by the interaction of upregulated miR133b with miR-151-5p [46]. Further, the indirect interaction of upregulated mature miR192-5p with miR-203 [47] predicted activation of the promoter of the surviving *BIRC5*. We also identified multiple upregulated miRNAs (miR-148a-3p, miR-133a-3p, miR-342-3p, and miR-373-3p/302) known to target *TNF* expression and associated adipogenesis, cell proliferation, and apoptosis [48,49,50]. In addition, this MAP analysis also showed an association of downregulated miR-216a-3p/128 cluster [42] with the activation of *ADORA2B* (Figure 3B,C).

### 3.3. Validation of Selected Differentially Expressed miRNAs in EVs Using Quantitative RT-PCR

Next, we validated the alterations in the levels of miRNAs in EVs from HIV-infected stimulant users with and without PH as identified by small RNA sequencing. For this, we selected 13 upregulated and 14 downregulated miRNAs based on criteria of absolute fold-change ≥ 1.5 and FDR ≤ 0.05 as seen in the volcano plot, common in analysis of miRNA gene interaction networks by IPA and gene enrichment analysis and significant relevance to disease pathology known in the literature and/or previously shown to be associated with HIV infection [51,52,53] or inflammation and oxidative stress. As shown in Figure 4A, significantly higher levels of miR6501, miR-203b, miR-32-5p, miR-373-3p, miR-133a-5p, miR-192-5p, miR-335-3p, and miR-133A-3p were observed in the plasma-derived EVs from the HIV + Stimulants + PH group compared to EVs from the HIV + Stimulants group while miR-4670-3p, miR-6778-3p, miR-5571-5p, miR-185-3p, miR-19a-3p (*p* < 0.05), miR-564, and miR-216 (Figure 4B) were found to be significantly low in HIV + Stimulants + PH group. Due to the limitation of EV-derived RNA samples, we were not able to validate additional upregulated miRNAs (miR-92b-3p, miR-301a-3p, miR-4458, and miR-5688). Nevertheless, they were investigated in the validation cohort as mentioned below.

### 3.4. Comparison of EV-miRNA Cargo in an Independent Group of HIV-Infected Cocaine Users with and without PH

The above mentioned findings using small RNA sequencing were on EVs isolated from HIV-infected poly-drug users, so we next validated the alterations in EV-linked miRNAs in a separate cohort with samples from HIV-infected cocaine users with and without PH. We first compared levels of selected miRNAs between uninfected (UI), uninfected individuals with cocaine use (Coc), HIV-infected individuals (HIV), and HIV-infected individuals with cocaine use (HIV + Coc) (*n* = 6/group) to see the effect of cocaine abuse in HIV-infected individuals. The levels of miRNA 32-5p, miR-192-5p, miR-335-3p, and miR-6501 were significantly higher in HIV-positive cocaine users compared to UI controls while no significant differences were observed in EVs from Coc or HIV groups (Figure 5A). However, a comparison of these significantly upregulated miRNAs in EVs from HIV-infected individuals with HIV-PH individuals (*n* = 12/group; *n* = 6 cocaine users, *n* = 6 non-drug users) revealed only miRNA 32-5p to further significantly increase in HIV-PH patients (Figure 5B), regardless of cocaine use. Notably, miR-92b-3p and miR-301a-3p were also significantly high in EVs from HIV + PH patients compared to HIV patients without PH, but this increase was associated with cocaine abuse (Figure 5C). This analysis confirmed sequencing analysis findings on HIV-infected poly-stimulant users showing higher levels of EV-linked miR-32-5p, miR-92-b-3p, miR-203-b, and miR-301-3p targeting cellular proliferation in HIV-PH patients.

Similarly, significantly downregulated miRNAs in EVs from HIV-infected stimulant users were also validated in an independent cohort of HIV-infected cocaine users with or without PH. The levels of miR-6859, miR-6778, and miR-5571 were found to be significantly downregulated in HIV-infected cocaine users or non-drug users when compared with uninfected non-drug users (Figure 6A). Comparison of these miRNAs between HIV-infected individuals with and without PH revealed significantly lower levels of miR-5571 along with miR-46708, miR-564, and miR-216a in HIV-PH patients (Figure 6B); however, this difference was not significantly associated with cocaine abuse (Figure 6C). Downward trends of miR-564 levels were observed in HIV-PH cocaine users compared to HIV-PH nondrug users.

Correlation analysis of significantly altered EV-linked miRNAs in HIV-PH patients with clinical parameters found a significant correlation of EV-linked miR-92b-3p (R^2^ 0.839, *p* < 0.001) and miR-301a-3p with CD4 (Figure 7A). In addition, higher levels of miR-301a-3p were also found to be correlated with viral load, DLCO, and FEV1/FVC ratio. Among the downregulated miRNAs, miR-4670 and miR-5571 were significantly correlated with CD4 and viral load (Figure 7B); however, miR-4670 was also co-related with FEV1/FVC (R^2^ 0.582, *p* = 0.004). Decreased levels of miR-564 in EVs only showed a significant correlation with DLCO (R^2^ 0.455, *p* = 0.023).

## 4. Discussion

In this study, we first compared the miRNA cargo of plasma-derived EVs from HIV-infected poly-drug users with and without PH using small RNA sequencing. Among the most elevated miRNAs found in EVs from PWH with PH were those implicated in stimulating cell proliferation and inhibiting intrinsic apoptotic signaling pathways. On the other hand, the downregulated miRNAs in the individuals with PH indicated a possible correlation with the signaling pathways mediated by transmembrane receptor protein tyrosine kinase, IL-2, and the negative regulation of smooth muscle cell proliferation. The limitation of this analysis was the small sample size used for RNA sequencing. Therefore, we next validated the identified significantly altered miRNAs in EVs from an independent set of 36 HIV-infected individuals with and without PH using quantitative RT-PCR. We confirmed increased levels of miR-32-5p, 92-b-3p, and 301a-3p positively regulating cellular proliferation and decreased levels of miR-5571, -4670, negatively regulating smooth muscle proliferation in EVs from HIV-PH patients. The FEV1/FVC ratio and CD4 count showed a negative correlation with the higher levels of miR-301a-3p and lower levels of miR-4670 in EVs from PWH, whereas the viral load showed a positive correlation. All of these findings suggest a possible link between HIV-PH and changes in the miRNA cargo of circulating EVs.

EVs are released into the peripheral circulatory system directly from the cell surface plasma membrane or from the fusion of intracellular multivesicular bodies with the cell surface membrane [54]. EVs’ ability to merge with and release their contents into cells that are different from their source cells, as well as their potential to affect processes in the recipient cells by transferring their protein or coding and non-coding RNA cargo including miRNAs, have been the subject of extensive research by many researchers [14,55,56,57]. In a study of cancer patients, Yuan T et al. found miRNAs as one of the most abundant plasma extracellular RNA species [58]. Although the sorting of miRNA into EVs is not completely understood, the mechanism could be dependent or independent of the miRNA sequence and involves binding with either RNA-binding proteins or EV biogenesis proteins [59,60]. Cocaine on binding to sigma-1 receptor (Sig-1R) has been shown to stimulate EV secretion from neuroblastoma cells by dissociating the Sig-1R from ARF6 (ADP-ribosylation factor 6) protein involved in EV trafficking [61]. Further, HIV proteins and cocaine are known to cause oxidative stress [62,63] that can alter the expression of miRNA in cells [9,64]. This may then alter the sorting of miRNAs into EVs.

The analysis using 12 PH datasets available in the GEO database showed significant enrichment of miRNAs related to the positive regulation of cell proliferation and negative regulation of intrinsic apoptosis signaling in EVs from HIV-infected stimulant users with pulmonary hypertension (HIV + Stimulants + PH). Interestingly, previous studies investigating the EV proteome identified increased levels of cell proliferative and anti-apoptotic-associated proteins in plasma-derived EVs from HIV-positive individuals that correlated with immune activation markers [65]. In the upregulated pathway enrichment analysis, the vast majority of miRNAs were related to *TGFBR2, TNF, BMPR2, NF2, HIF1A*, and *KLF4* genes in the regulation of cell proliferative pathways (miR-373-3p, miR-9-5p, miR-148a-3p, miR-92b-3p, and miR-301a-3p) and *BCL2, MMP9, BCL2L1, BCL2L2* in the negative regulation of intrinsic apoptotic signaling pathway (miR-133a-3p, miR-451a, miR-373-3p, miR-9-5p, miR-192-5p, miR-148a-3p, and miR-203b-3p). Previous studies have suggested that miR-203b-3p suppresses *Bcl-xL* protein expression via direct binding to the gene’s mRNA 3’UTR and correlates negatively with *BCL2L1* mRNA expression in breast tumors [66]. Alternatively, the downregulated miRNAs in the HIV + Stimulants + PH group were associated with *IL-2RA* and *IL-2RB* in interleukin-2-mediated signaling (miR-6778), with *ERBB3* and *DDR1* in the transmembrane receptor protein tyrosine kinase signaling (miR-6778 and 185-3p) and *HMOX1, PPARG, IGFBP3*, and *BMPR2* targets involved in the negative regulation of smooth muscle cell proliferation (miR216a-3p, miR-19b-3p). Additional IPA analysis predicted the inhibition of *BMPR2* by upregulated miR-32-5p/miR-92b-3p in EVs from the HIV + Stimulants + PH group. In addition, it predicted activation of the *TNFSF10 (TRAIL)* by upregulated miR133b, activation of survivin oncogene (*BIRC5*) by upregulated miR192-5p, and activation of adenosine receptor, *ADORA2B* by the downregulated miR-216a-3p/128 cluster. Together, these changes are consistent with a signature of hyperproliferative SMC, which is believed to be involved in PH development.

Aliotta et al. studied the pathological role of EVs in the mice model of monocrotaline-induced PAH [67,68]. They found that small-sized EVs from the lung and plasma of monocrotaline PH mice can develop PAH in healthy mice. This was associated with higher levels of pro-proliferative miRNAs in small EVs such as miR-19b, miR-20a, miR-20b, and -145 targeting *BMPR 2* signaling. Further, they also reported increased levels of these miRNAs in EVs from idiopathic PAH patients in addition to other unique miRNAs including miR-148a [67] which we also observed to be high in EVs from HIV-PH patients.

Our previous studies identified higher numbers of plasma-derived EVs carrying higher levels of *TGF-β1* in HIV-infected individuals (cocaine users and non-drug abusers) with PH compared to those without PH and this was found to be associated with pulmonary endothelial injury and smooth muscle hyperplasia, again two key components of PH development [18,69,70]. Validation studies using this same set of EVs from HIV-infected cocaine users confirmed the upregulation of miR-32-5p, -92-b-3p, and -301a-3p known to positively regulate cellular proliferation and downregulation of miR-5571, -4670 negatively regulating smooth muscle proliferation in EVs from HIV-PH patients. Interestingly, an increase in the levels of EV-linked miR-301a-3p and a decrease in miR-4670 in HIV-PH patients positively correlated with viral load and negatively with CD4 count, and FEV1/FVC ratio. Whether this could be related to (a) adherence to HAART and (b) stimulant consumption-associated development of COPD with PH could be a potential follow-up study.

In addition to the above-discussed miRNA changes with features of PH development, multiple further lines of evidence support the identified miRNA changes in EVs as functional markers of PH. For instance, it has been demonstrated that the miR-130/301 family of miRNAs is produced by a variety of factors that cause pulmonary hypertension, and inhibiting the action of these miRNAs reversed the disease in pre-clinical animal models of PH [5,71]. In our previous study, we reported increased levels of miR-301a in cocaine and HIV-Tat-treated human pulmonary arterial smooth muscle cells (HPASMC) and its involvement in the post-translational regulation of *BMPR2* expression leading to enhanced smooth muscle cell proliferation. While mutations in the *BMPR-2* have been linked to familial pulmonary arterial hypertension, multiple studies indicate decreased *BMPR* expression irrespective of mutations in the gene [72,73] contributing to the pathogenesis of PH. In another study, we observed that higher levels of miR-130a in the EVs derived from HIV-infected cocaine-treated macrophages were associated with the decrease in the expression of *PTEN* (phosphatase and tensin homolog and tuberous sclerosis 1 and 2) and activation of proliferative *PI3K/AKT* signaling in pulmonary arterial smooth muscles cells on treatment with these macrophage-derived EVs [5]. The same mechanisms could be associated with EVs loaded with miR 301a in HIV-PH subjects, as found in the current study, which has the potential to be used as an effective diagnostic biomarker. Additionally, we previously identified miR-216 to be upregulated in hyperproliferative cocaine and HIV-Tat-treated HPASMCs and associated negative regulation of *BMPR2* translation [9]. However, miR-216 was found to be downregulated in plasma-derived EVs possibly due to the differential sorting mechanism of miRNAs in EVs [59] and/or given the fact that EVs in peripheral blood are expected to be derived from many cell types.

Multiple research studies demonstrate that the proliferative *TGFβ1* arm of the *TGFβ* superfamily signaling can be stimulated by the failing anti-proliferative *BMPR*-axis, which can result in excessive smooth muscle cell proliferation and PH [10,11,74]. The EV-linked miR-192-5p found to be increased in HIV-PH patients mediates *TGF-β/Smad 3*-induced renal fibrosis [75]. In addition, miR-564 was consistently found to be downregulated in EVs from HIV-PH patients targeting *TGF-β1* and associated with the reduced proliferation and invasion of glioblastoma cells [76]. miR-564 has been identified as a dual inhibitor of *PI3K* and *MAPK* pathways and reducing cell proliferation in breast cancer through G1 arrest [77]. The miR-185 which can negatively modulate *TGF-β1* expression [78] was also observed to be at low levels in the EVs from HIV-PH patients compared to non-PH HIV-infected individuals which could potentially lead to increases in the *TGF-β1*-dependent pulmonary endothelial injury and smooth muscle hyperplasia [18].

Although aberrant *TGF-β/BMPR2* signaling is most commonly associated with the development of PH, other factors can also cause endothelial smooth muscle cell dysfunction. These factors may include mitochondrial damage and oxidative stress [79,80]. In the early phases of pulmonary artery hypertension, oxidative stress-mediated vasoconstriction is likely one of the most important contributors [81,82]. Past research on HIV patients found alterations in plasma EV cargo, including RNA linked to oxidative stress, inflammation, and persistent immune activation [83,84]. EV-associated miRNAs, miR-6501, miR-373, miR-133, miR-192, and miR-335 were increased in HIV-Stim-PAH subjects which previously have been shown to correlate with oxidative stress. miR-373 has been previously shown to be elevated upon infection with HIV-1 [52], whereas miR-133 is known to play a role in vascular stress, remodeling, and cell survival [34] and previous study from our lab has also shown that lncRNA known to target miRNA-133 was elevated in HPASMC on the treatment of cocaine and HIV-Tat [85]. Also, the literature suggests that miR-133 and other miRNAs including miR-373 are increased in vascular tissues during inflammation and oxidative stress [86,87,88,89].

In summary, our findings indicate alterations in the levels of miRNAs associated with promoting cell proliferation and suppressing intrinsic apoptotic signaling pathways in small EVs from the plasma of HIV-drug users with PH (Figure 8). To our knowledge, this is the first report on the analysis of non-coding RNA cargo in circulating EVs from HIV-infected individuals with PH. The independent validation of circulating EVs isolated from HIV-infected cocaine users and nondrug users with and without PH shows the association of higher levels of EV-linked miR-32-5p, 92-b-3p, and 301a-3p in HIV-PH patients with the positive regulation of cellular proliferation and lower levels of miR-5571, -4670 in EVs with the negative regulation of smooth muscle proliferation. Collectively, these data suggest that alterations in the levels of these miRNAs in circulating EVs are associated with the development of PH in HIV-drug users. Further studies in larger cohorts are needed to confirm the identified EV-linked miRNA markers associated with PH in PLWH, to find the cellular origin of the significantly altered miRNA-linked EVs, and to assess whether target genes of identified altered miRNAs are up- or downregulated in vascular cells of remodeled vessels by in situ hybridization of lung sections or total RNA analysis of lungs from HIV-PH patients. Such work would provide a better understanding of the role of EV miRNAs in HIV-PH pathogenesis and identify a potential circulating prognostic and diagnostic biomarker of PH in PLWH.

## Figures and Tables

**Figure 1 cells-13-00886-f001:**
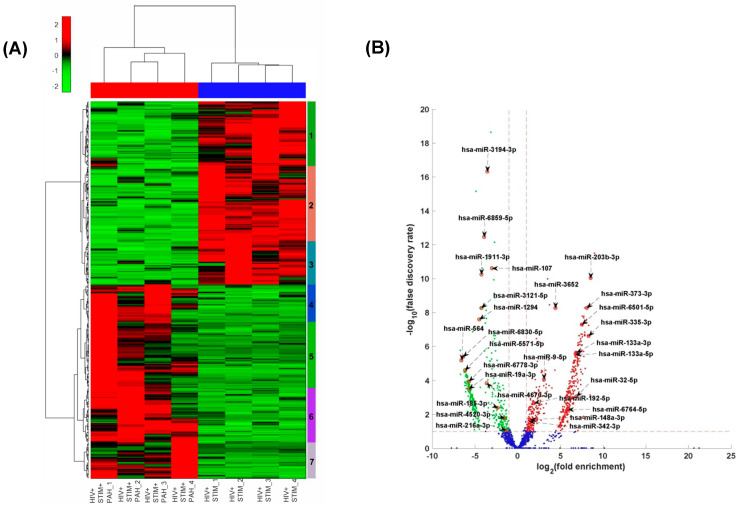
(**A**) Heatmap of the standardized expression of miRNAs that are significantly differentially expressed between HIV + STIM + PAH and HIV + STIM samples. MiRNAs are represented in rows and samples in columns. The normalized expression data were row standardized (zero mean and unit variance) with negative values in green representing relatively low expression and positive red values representing relatively high expression. The data were hierarchically clustered both row and column wise using the Euclidean distance measure and Ward’s linkage function. (**B**) Volcano plot of the differential expression profile of miRNAs between HIV + STIM + PAH and HIV + STIM samples. The x-axis represents the log of the differential expression ratio, and the y-axis represents the negative log of the false discovery rate. The vertical red perforated lines represent the +/− 1.5-fold-change values. The horizontal red perforated line represents the 0.05 FDR value. MiRNAs that are significantly downregulated in HIV + STIM + PAH relative to HIV + STIM are shown in green and those that are significantly upregulated are shown in red.

**Figure 2 cells-13-00886-f002:**
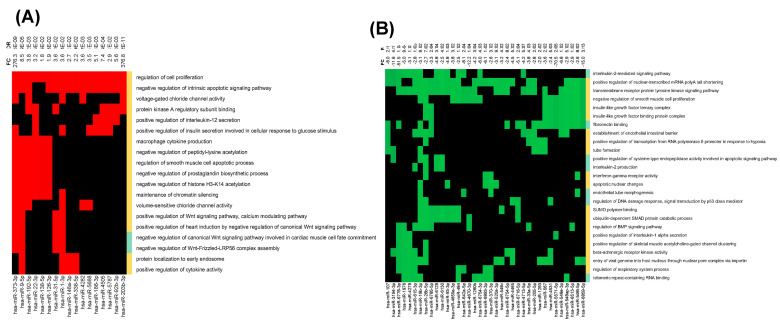
Over-representation analysis using miEAA. (**A**) Significantly upregulated and (**B**) significantly downregulated miRNAs depicted as a heatmap representing the effected pathways. The diagrams only represent pathways that contain at least four miRNAs and miRNAs that are in at least four pathways. The cells corresponding to an miRNA that is associated with regulated pathways are colored red for upregulated and green for downregulated pathways whereas the not significantly altered pathways are in black. The expression fold difference of the miRNAs is given on top of the heatmap along the column of the corresponding miRNA. The color-bar along the right margin of the heatmap indicates the over (orange)- or under (green)-representation status of the miRNAs in the pathway.

**Figure 3 cells-13-00886-f003:**
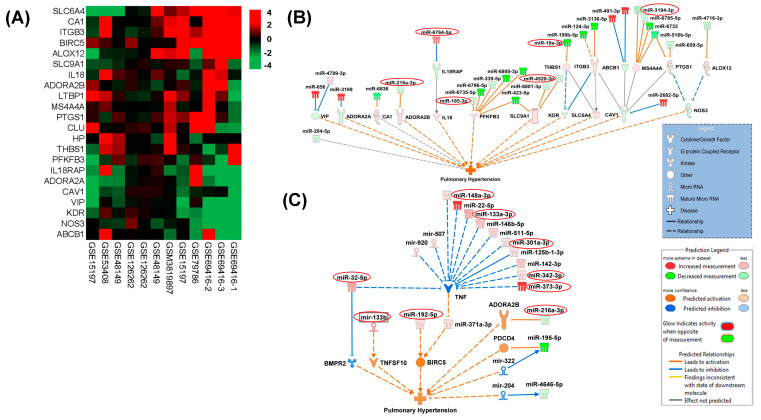
(**A**) Hierarchically clustered heatmap of differential gene expression between PAH and Control samples in 12 GEO datasets. The log ratio of genes that showed consistent differential expression changes across the 12 samples are shown in the heatmap with red indicating up-regulation and green indicating down-regulation. These genes were selected for their direct association with pulmonary hypertension based on the literature evidence. (**B**,**C**) Shown are molecular interaction networks between the significantly differentially expressed miRNAs (HIV + STIM + PAH vs. HIV + STIM) and genes associated with pulmonary hypertension leading to a predicted activation state of the function as deduced by IPA’s causal analysis tools. Solid lines indicate direct molecular interactions and perforated lines indicate indirect interactions.

**Figure 4 cells-13-00886-f004:**
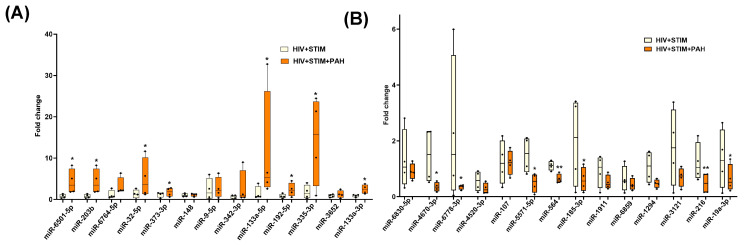
Quantitative RT-PCR analysis of some of the significantly upregulated (**A**) and downregulated miRNAs (**B**) in the plasma-derived EVs from HIV + Stim + PH group compared to EVs from HIV + Stim group. The miRNAs were selected based on criteria of absolute fold-change ≥ 1.5 and *p*-value ≤ 0.05 as seen in volcano plot, common in the analysis of miRNA gene interaction networks by IPA and gene enrichment analysis; and significant relevance to disease pathology known in the literature and/or previously shown to be associated with HIV infection. * *p* < 0.05, ** *p* < 0.01 vs. HIV + STIM.

**Figure 5 cells-13-00886-f005:**
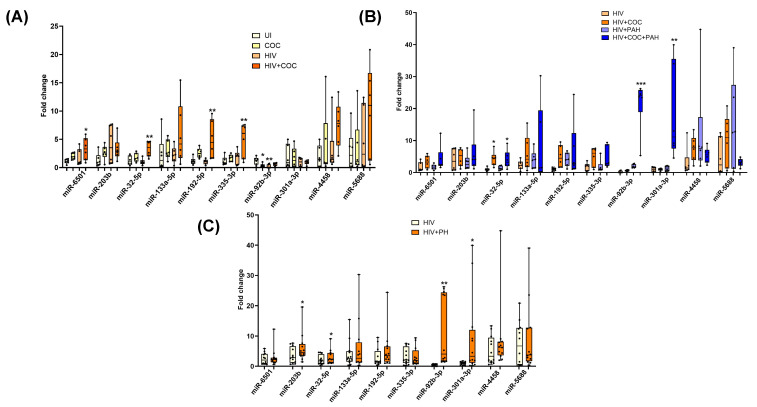
RT-PCR validation of significantly upregulated miRNAs in plasma-derived EVs from an independent cohort from the University of Pittsburgh. (**A**) The comparison between plasma-derived EVs from uninfected (UI), uninfected individuals with cocaine use (Coc), HIV-infected individuals (HIV), and HIV-infected individuals with cocaine use (HIV + Coc) (*n* = 6/group) * *p* < 0.05, ** *p* < 0.01 vs. UI and (**B**) compares EVs from HIV-infected individuals with and without PH individuals (*n* = 6/group) based on cocaine use * *p* < 0.05, ** *p* < 0.01, *** *p* < 0.001 vs. HIV while (**C**) shows overall comparison between the two groups (*n* = 12/group) * *p* < 0.05, ** *p* < 0.01 vs. HIV.

**Figure 6 cells-13-00886-f006:**
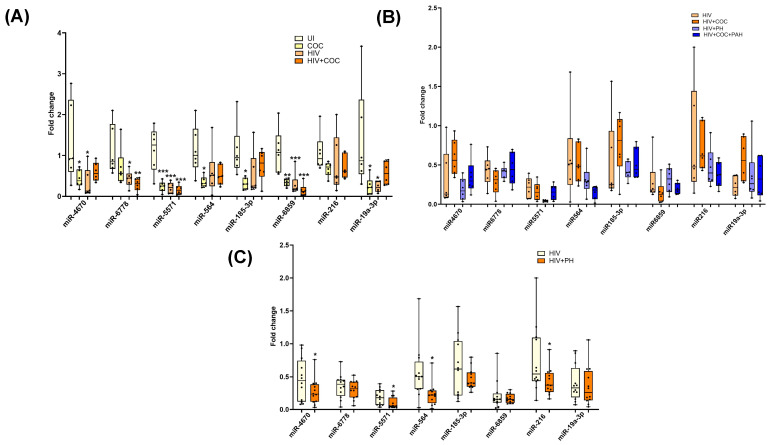
RT-PCR validation of selected significantly downregulated miRNAs in EVs from the University of Pittsburgh cohort. (**A**) Comparison of plasma-derived EVs from uninfected (UI), uninfected individuals with cocaine use (Coc), HIV-infected individuals (HIV), and HIV-infected individuals with cocaine use (HIV + Coc) (*n* = 6/group) * *p* < 0.05, ** *p* < 0.01, *** *p* < 0.001 vs. UI. (**B**,**C**) The levels of downregulated miRNA in EVs from HIV-infected individuals with and without PH individuals showing drug usage (*n* = 6/group) (**C**) and overall comparison (*n* = 12/group). * *p* < 0.05 vs. HIV.

**Figure 7 cells-13-00886-f007:**
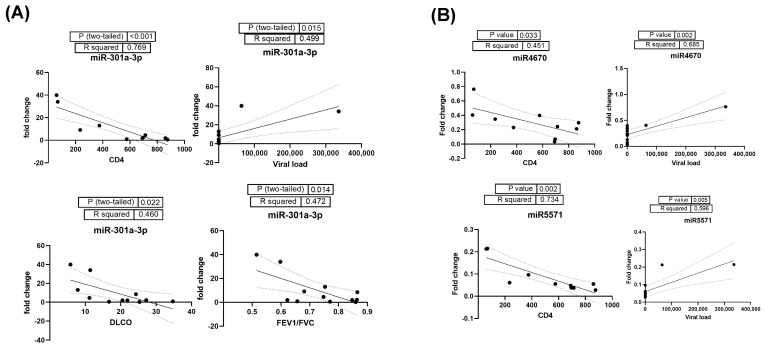
(**A**) Correlation analysis of significantly upregulated EV-linked miR-301a-3p in HIV-PH patients with clinical parameters CD4, viral load, DLCO, and FEV1/FVC ratio. (**B**) Correlation analysis of HIV-PH associated downregulated miR-4670 and miR-5571 with CD4 and viral load in HIV-infected individuals.

**Figure 8 cells-13-00886-f008:**
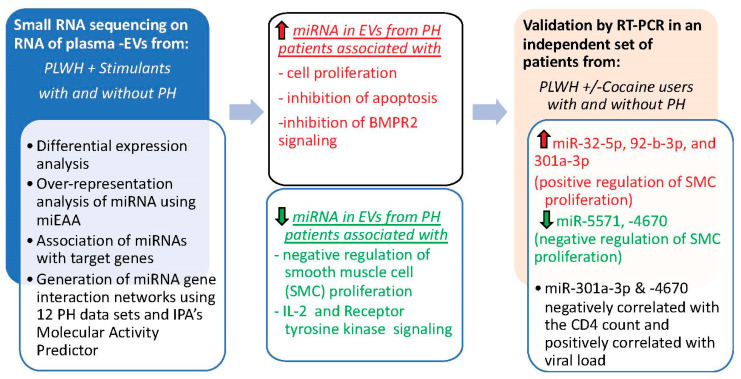
Schematic summarizing the miRNA analysis of EVs from people living with HIV with and without pulmonary hypertension. Upregulated molecules are colored red and downregulated molecules are colored green.

**Table 1 cells-13-00886-t001:** Selected miRNAs in the seven sub-clusters of the dendrogram in Figure 1A formed by hierarchically clustering the miRNA and their associated biological functions.

HIV + STIM + PH Downregulated	Interleukin-2-Mediated Signaling Pathway GO0038110	Negative Regulation of Smooth Muscle Cell Proliferation GO0048662	Transmembrane Receptor Protein Tyrosine Kinase Signaling Pathway GO0007169	SUMO Polymer Binding GO0032184
Cluster 1	hsa-miR-19a-3p, hsa-miR-484, hsa-miR-6130, hsa-miR-642a-5p	hsa-miR-19a-3p, hsa-miR-26b-5p, hsa-miR-484, hsa-miR-642a-5p	hsa-miR-19a-3p, hsa-miR-26b-5p, hsa-miR-484, hsa-miR-6130, hsa-miR-642a-5p	hsa-miR-19a-3p, hsa-miR-26b-5p, hsa-miR-484, hsa-miR-6130
Cluster 2	hsa-miR-615-3p, hsa-miR-6754-3p, hsa-miR-6778-3p	hsa-miR-1260b, hsa-miR-4659a-3p, hsa-miR-525-5p, hsa-miR-548e-3p, hsa-miR-6754-3p, hsa-miR-6859-5p	hsa-miR-1260b, hsa-miR-4520-3p, hsa-miR-4530, hsa-miR-4659a-3p, hsa-miR-525-5p, hsa-miR-548e-3p, hsa-miR-548v, hsa-miR-615-3p, hsa-miR-6778-3p, hsa-miR-6785-5p, hsa-miR-6794-3p, hsa-miR-6809-5p, hsa-miR-6859-5p	hsa-miR-4659a-3p, hsa-miR-548v, hsa-miR-6785-5p
Cluster 3	hsa-miR-1294, hsa-miR-4279, hsa-miR-4763-5p, hsa-miR-6129, hsa-miR-6855-3p, hsa-miR-93-3p		hsa-miR-1207-5p, hsa-miR-1294, hsa-miR-1911-3p, hsa-miR-4279, hsa-miR-4763-5p, hsa-miR-567, hsa-miR-6129, hsa-miR-6796-5p, hsa-miR-6855-3p, hsa-miR-93-3p	hsa-miR-1911-3p, hsa-miR-6129, hsa-miR-6796-5p, hsa-miR-93-3p
**HIV + STIM + PH** **Up Regulated**	**Regulation of Cell Proliferation GO0042127**		**Negative Regulation of Intrinsic Apoptotic Signaling Pathway GO2001243**	**Positive Regulation of Interleukin-12 Secretion GO2001184**
Cluster 4	hsa-miR-1-3p, hsa-miR-126-3p, hsa-miR-342-3p, hsa-miR-9-5p		hsa-miR-1-3p, hsa-miR-126-3p, hsa-miR-342-3p, hsa-miR-9-5p	
Cluster 5	hsa-miR-1224-5p, hsa-miR-1292-3p, hsa-miR-186-3p, hsa-miR-22-3p, hsa-miR-335-3p, hsa-miR-3652, hsa-miR-373-3p, hsa-miR-4282, hsa-miR-5787, hsa-miR-660-3p, hsa-miR-92b-3p		hsa-miR-1224-5p, hsa-miR-1292-3p, hsa-miR-186-3p, hsa-miR-3652, hsa-miR-373-3p, hsa-miR-4282, hsa-miR-5787, hsa-miR-629-3p, hsa-miR-660-3p, hsa-miR-92b-3p	hsa-miR-186-3p, hsa-miR-22-3p, hsa-miR-335-3p, hsa-miR-5787, hsa-miR-629-3p, hsa-miR-92b-3p
Cluster 6	hsa-miR-144-3p, hsa-miR-148a-3p, hsa-miR-149-3p, hsa-miR-192-5p, hsa-miR-302a-5p, hsa-miR-31-5p, hsa-miR-3672, hsa-miR-4505, hsa-miR-4733-5p, hsa-miR-548f-3p, hsa-miR-5688, hsa-miR-579-3p, hsa-miR-6501-5p, hsa-miR-6797-5p, hsa-miR-7844-5p		hsa-miR-144-3p, hsa-miR-148a-3p, hsa-miR-149-3p, hsa-miR-192-5p, hsa-miR-302a-5p, hsa-miR-31-5p, hsa-miR-4429, hsa-miR-4505, hsa-miR-4733-5p, hsa-miR-548f-3p, hsa-miR-5688, hsa-miR-6797-5p, hsa-miR-7844-5p	hsa-miR-3672, hsa-miR-4429, hsa-miR-4505, hsa-miR-579-3p, hsa-miR-6501-5p
Cluster 7	hsa-miR-138-5p, hsa-miR-203b-3p, hsa-miR-320d, hsa-miR-338-5p, hsa-miR-3690, hsa-miR-4458		hsa-miR-203b-3p, hsa-miR-320d, hsa-miR-338-5p, hsa-miR-3690, hsa-miR-4458	

## Data Availability

Data will be available upon a reasonable request.

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
