# Peer review of "Analysis of MicroRNA Cargo in Circulating Extracellular Vesicles from HIV-Infected Individuals with Pulmonary Hypertension"

_cells, 2024, doi:10.3390/cells13110886_

Round 1

Reviewer 1 Report

Comments and Suggestions for Authors

Please include a paragraph after the discussion summarizing the study findings and also add a section on conclusions.

A figure illustrating the study findings will greatly improve the clarity of the study and will help the readers to gain a better understanding of the study outcomes.

Author Response

1-Please include a paragraph after the discussion summarizing the study findings and also add a section on conclusions.

Response: We have now included a conclusion paragraph  in the end of discussion.

 2-A figure illustrating the study findings will greatly improve the clarity of the study and will help the readers to gain a better understanding of the study outcomes.

Response:  Thank you for this excellent suggestion. This is now included as Figure 8.

Reviewer 2 Report

Comments and Suggestions for Authors

Pulmonary hypertension poses a significant comorbidity challenge for people living with HIV (PLWH). This study explores the miRNA composition within plasma-derived extracellular vesicles (EVs) from PLWH, comparing those with and without pulmonary hypertension (PH). The findings reveal intriguing data, indicating that EVs from PLWH with PH contain miRNAs associated with promoting cell proliferation and suppressing intrinsic apoptotic signalling pathways. This research underscores the correlation between alterations in miRNA cargo within circulating EVs and PH in PLWH, advancing our comprehension of the pathophysiological mechanisms underlying smooth muscle proliferation linked to PH. Furthermore, the independent validation of multiple microRNAs strengthens the study's findings and sets the stage for identifying functional biomarkers of the disease.

Author Response

Pulmonary hypertension poses a significant comorbidity challenge for people living with HIV (PLWH). This study explores the miRNA composition within plasma-derived extracellular vesicles (EVs) from PLWH, comparing those with and without pulmonary hypertension (PH). The findings reveal intriguing data, indicating that EVs from PLWH with PH contain miRNAs associated with promoting cell proliferation and suppressing intrinsic apoptotic signaling pathways. This research underscores the correlation between alterations in miRNA cargo within circulating EVs and PH in PLWH, advancing our comprehension of the pathophysiological mechanisms underlying smooth muscle proliferation linked to PH. Furthermore, the independent validation of multiple microRNAs strengthens the study's findings and sets the stage for identifying functional biomarkers of the disease.

Response: We appreciate positive and encouraging comments from the reviewer.

Reviewer 3 Report

Comments and Suggestions for Authors

As a follow-up study of their previous work, Aatish Mahajan et al. investigated a correlation between microRNA cargo in circulating Exsomes PLWH and pulmonary hypertension in the current study. They found that the expression of miR-301a-3p and  miR-4670 is associated with the disease as an observation. The study is well-designed, and miR profiles are analyzed using multiple bioinformatics tools; however, the reviewer suggests that the current study should be modified by providing additional data.

They need more explanation on how/why they decided to focus on the miRNA profile. Generally, non-coding RNAs (ncRNAs) are classified into two groups: small/ short ncRNAs and long ncRNAs, and the authors focused on miRNAs. They need to provide more background on why they decided to concentrate on miRNAs rather than long ncRNAs.
Each miRNA is produced via Dicer and Drosha from transcribed immature RNAs. They reported that the expression of miR-301a-3p and  miR-4670 changed in the PLWH cohorts. Has the expression of each gene containing premiR of each miRNA also changed accordingly? How is the host gene activation regulated? Is the regulatory mechanism controlled by opioid receptor-mediated signals?

What mechanism is involved in the incorporation of miRNA in EV? As they described before, miRNA and lncRNA can interact. Is some lncRNA associated with the incorporation?

They address the fact that the miRNA identifies a potential circulating prognostic and diagnostic biomarker of PH in PLWH. Is there any correlation between virus load and the miRNA expression?

Author Response

1- They need more explanation on how/why they decided to focus on the miRNA profile. Generally, non-coding RNAs (ncRNAs) are classified into two groups: small/ short ncRNAs and long ncRNAs, and the authors focused on miRNAs. They need to provide more background on why they decided to concentrate on miRNAs rather than long ncRNAs.
Each miRNA is produced via Dicer and Drosha from transcribed immature RNAs. They reported that the expression of miR-301a-3p and  miR-4670 changed in the PLWH cohorts. Has the expression of each gene containing premiR of each miRNA also changed accordingly? How is the host gene activation regulated? Is the regulatory mechanism controlled by opioid receptor-mediated signals?
Response: We thank reviewer for this suggestion and now have included the details on other noncoding RNAs in EVs and also why we were interested in examining EV linked  miRNA cargo from HIV-infected individuals with cardio-pulmonary complications. In this manuscript we didn’t examine if pre-miRNA for each miRNA was also present in EVs. To identify potential candidate PH-associated gene targets for altered miRNAs we did  IPA analysis using 12 datasets, available in the GEO database associated with pulmonary hypertension. We also used IPA’s Molecule Activity Predictor (MAP) to predict the activated state of pulmonary hypertension explained by the differential expression pattern of miRNAs in our data and the expression pattern of their known (literature based ) associated target genes. In response to the reviewer comment ,  we have now included how cocaine receptor ( sigma -1 receptor) may be involved in regulating the release of EVs.

2-What mechanism is involved in the incorporation of miRNA in EV? As they described before, miRNA and lncRNA can interact. Is some lncRNA associated with the incorporation?

Response: We have now included the details on sorting of miRNA in EVs in the discussion.

3-They address the fact that the miRNA identifies a potential circulating prognostic and diagnostic biomarker of PH in PLWH. Is there any correlation between virus load and the miRNA expression?

Response: Yes, we did see positive correlation of significantly altered miR-301a-3p and miR-4670  with viral load in PLWH as shown in figure 7.

Reviewer 4 Report

Comments and Suggestions for Authors

Dear authors, the manuscript is interesting and have a high scientific relevance. However, I have some concerns that might be helpful for improving the manuscript.

First, please specifically consider the mechanisms that can be involved in the development of pulmonary hypertension if cocaine is not taken by a person. These mechanisms should be considered in comparison with the mechanisms of pulmonary hypertension for patients that do no not take cocaine.

Then, please indicate whether patients were co-infected with hepatitis C or whether they have any other co-morbidities. 
How many samples are there in any group (HIV-uninfected, HIV-uninfected cocaine users etc.).

Please, compare the obtained results with other results known from literature or discussed in scientific studies earlier.

Comments on the Quality of English Language

Please, check English with the help of native speaker.

Author Response

1- First, please specifically consider the mechanisms that can be involved in the development of pulmonary hypertension if cocaine is not taken by a person. These mechanisms should be considered in comparison with the mechanisms of pulmonary hypertension for patients that do not take cocaine.

Response: We thank the reviewer for this suggestion and have now included this in the revised manuscript in the introduction and discussion.

2- Then, please indicate whether patients were co-infected with hepatitis C or whether they have any other co-morbidities. 

Response: These details are now included under theHuman samples and Data collection’ sub-section.

3- How many samples are there in any group (HIV-uninfected, HIV-uninfected cocaine users etc.).

Response: Thanks for pointing this out. We have now included the details: ‘ HIV-Uninfected non-drug users (n=6), HIV-uninfected Uninfected cocaine users (Coc) (n=6), people living with HIV non-drug users (PLWH) (n=6), PLWH cocaine users (PLWH + Coc) (n=6), PLWH with pulmonary hypertension (PLWH+PH) (n=12, n=6 non-drug users and n=6 cocaine users)’.

4- Please, compare the obtained results with other results known from literature or discussed in scientific studies earlier. 

Response: We have now added other related findings from the literature in the Discussion section.

Round 2

Reviewer 3 Report

Comments and Suggestions for Authors

The authors clearly have clearly explained the reviewer's comment. 

Reviewer 4 Report

Comments and Suggestions for Authors

Dear authors, the manuscript has been corrected according to my comments and suggestions. After careful check of English, I recommend to accept it for publication.

Comments on the Quality of English Language

English should be revised.